# Range aware message drop policy for delay tolerant networks

Samiullah Khan[1], Khalid Saeed[1], Muhammad Faran Majeed[2], Khursheed Aurangzeb[3], Zahoor Ahmad[1], Muhammad Shahid Anwar[4] and Piratdin Allayarov[5]

[1] Department of Computer Science, Shaheed Benazir Bhutto University Sheringal, Upper Dir, Khyber Pakhtunkhwa, Pakistan
[2] Department of Computer Science, Kohsar University Murree, Murree, Punjab, Pakistan
[3] Department of Computer Engineering, College of Computer and Information Sciences, King Saud University, Riyadh, Saudi Arabia
[4] Department of AI and Software, Gachon University, Seongnam-si, Republic of South Korea
[5] Department of Mathematical Methods in Economics, Faculty of Digital Economy, Tashkent State Economic University, Tashkent, Uzbekistan

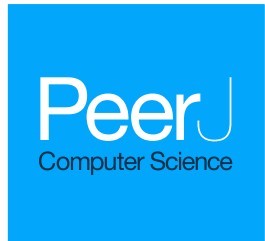

## ABSTRACT

In delay tolerant networks (DTNs) the messages are often not delivered to the destination due to a lack of end-to-end connectivity. In such cases, the messages are stored in the buffer for a long time and are transmitted when the nodes come into the range of each other. The buffer size of each node has a limited capacity, and it cannot accommodate the new incoming message when the buffer memory is full, and as a result network congestion occurs. This leads to a low delivery probability and thus increases the overhead ratio. In this research work, a new buffer management scheme called Range Aware Drop (RAD) is proposed which considers metrics such as message size and time to live (TTL). RAD utilizes TTL as an important metric and as a result, reduces the unnecessary message drop. Simulation results reveal that RAD performs significantly better than drop oldest (DOA) and size aware drop (SAD) in terms of delivery probability and overhead ratio. The obtained results also revealed that the hop-count average of SAD is 3.9 and DOA is 3.4 while the hop-count average of RAD is just 1.7. Also, the message drop ratio of the RAD is 36.2% while SAD and DOA have message drop ratios of 73.3% and 84.9% respectively.

## INTRODUCTION

Delay tolerant networks (DTNs) were first used in terrestrial networks having no direct connection all the time between sender and receiver nodes (*Perumal et al., 2022*; *Majeed et al., 2017a*). There are myriad factors for such disruption or disconnection *i.e.*, node movement, density, and limited radio range. The traffic is routed from source to destination *via* a store-carry-forward approach instead of TCP/IP. The store-carry and forward strategy is used to store the message in the buffer and transmit that message to the destination when communication is established between sender and receiver (*Gantayat & Jena, 2015*). However, this mechanism can lead to buffer congestion when several messages are stored in the buffer and there is no more space to accommodate the new incoming messages (*Hasan et al., 2023*).

Corresponding authors
Muhammad Faran Majeed,
m.faran.majeed@kum.edu.pk
Muhammad Shahid Anwar,
shahidanwar786@gachon.ac.kr

The major applications of DTNs are rural communication, studies of wild zebras, military and intelligence, and commercial purposes (*Kamal & Singh 2023*; *Majeed, Ahmed & Dailey, 2016*; *Verma, Savita & Kumar, 2021*; *Rashid et al., 2013a*). The architecture of DTNs is an overlay of existing networks which are divided into homogeneous regions. When a stream of data is transferred from the sender node to the receiver then it selects the appropriate path to deliver that message. If there is no such direct communication between the sender and receiver, then the store-carry-forward approach is used to store the data in the network until the data safely reaches the destination. To make the DTNs secure, the infrastructure of DTNs is protected from unauthorized users through different security mechanisms (*Majeed et al., 2017b*; *Rashid et al., 2013a*; *Rahman et al., 2024*). The intermittency and high-rate error in DTNs is reflected in Fig. 1.

The DTNs work on the concept of bundle layer and store-carry-forward mechanism. The bundle consists of different information such as source node, user data, data control information, and a bundle header. This layer is linked with TCP/IP to offer a gateway when contact is established between two nodes (*Verma, Savita & Kumar, 2021*; *Rashid et al., 2013a*; *LaFuente et al., 2024*). The concept of bundle layer and store-carry-forward mechanism in DTNs are shown in Figs. 2 and 3, respectively (*Gantayat & Jena, 2015*; *Khan et al., 2023*).

Communication in DTNs is done with the help of routing techniques. Routing means finding a good path to a known destination keeping in view the resource constraints. Different routing techniques used in DTNs are epidemic routing, PROPHET routing, spray & wait routing, single hop routing, and source routing (*Abubakar et al., 2020*; *Mangrulkar & Atique, 2010*; *Li et al., 2024*). All these are traditional routing techniques holding some basic information while many AI-based routing techniques properly show the bundle of messages that have been transferred to the destination. These routing techniques are based on machine learning (ML) algorithms. The overhead ratio of AI-based routing techniques is always high due to the usage of additional resources to classify the message into arrived and not arrived categories (*Gao & Zhang, 2023*; *Tekouabou, Maleh & Nayyar, 2022*; *Khattak et al., 2020*).

## Buffer management in DTNs

In DTNs, each buffer has a limited size and message congestion occurs when the buffer overflows (*Sobin, 2016*). To solve this problem, different buffer management policies are used to accommodate the new incoming message. Some of the buffer management policies in DTNs are as follows (*Rashid & Ayub, 2023*).

**Drop random policy:** This policy drops the messages randomly without network knowledge to accommodate the new message in the buffer. The messages are dropped from the buffer to store the incoming message. It drops the messages from the buffer so that sufficient space is created to hold the new arriving message in the buffer. The main objective of this mechanism is to improve the delivery probability in *ad-hoc* networks (*Mangrulkar & Atique, 2010*; *Majeed et al., 2021*).

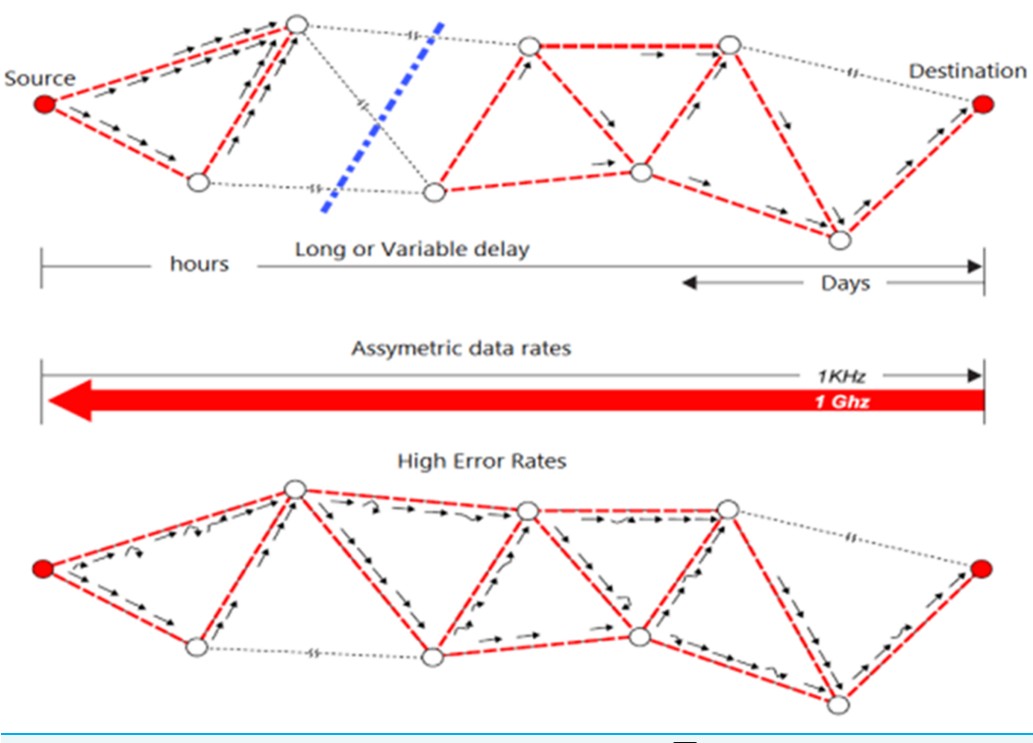

**Figure 1  Intermittency and high error rate in DTNs.**

**Drop oldest (DOA)/shortest lifetime first policy:** This policy focuses on dropping a message that has the shortest time to live (TTL) in the network. Such a message has a high chance of being delivered in the past to its destination as its TTL is low. This policy stores a message in the buffer by removing the lowest TTL message from the buffer. It increases the delivery probability of new messages (*Mangrulkar & Atique, 2010*; *Hasan et al., 2022*).

**E-drop (effective) policy:** In the E-drop policy, a message with all relevant information from the buffer with an equal or nearly equal size to the incoming message would be dropped. It accommodates an incoming message easily if the equal or nearly size message from the buffer is dropped. It minimizes the unnecessary message drop due to the selection of appropriate size messages. The messages smaller than incoming messages have a large opportunity to be delivered because they remain for a longer time in the buffer (*Rashid et al., 2011*; *Saeed et al., 2021*).

**Size aware drop buffer management scheme:** Size aware drop (SAD) is used to increase buffer utilization and avoid unnecessary message drops by discarding the most appropriate message from the buffer. It randomly selects the messages from the buffer. It also selects multiple small messages to create space for incoming messages in case of having no equal or at least large messages in the buffer. It avoids unnecessary message drops due to picking the appropriate size message from the buffer. It maximizes the delivery probability and minimizes the overhead ratio of the messages (*Rehman et al., 2021*).

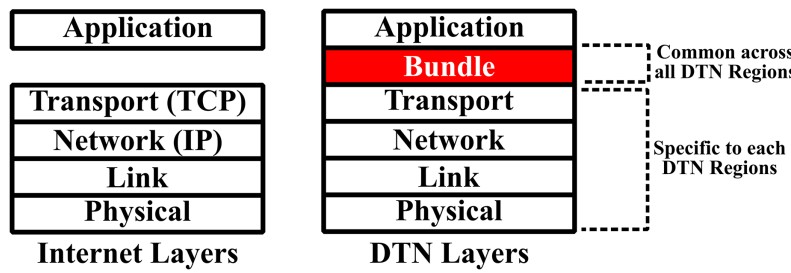

**Figure 2** Bundle layer *Rehman et al. (2021)*.

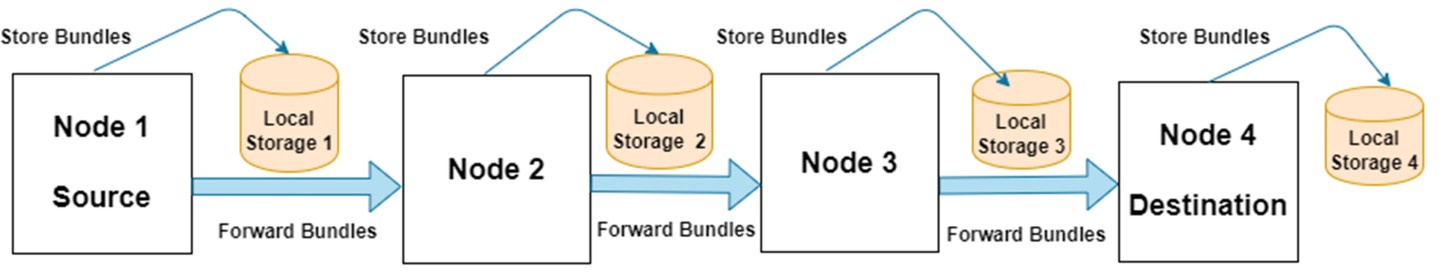

**Figure 3** Store-carry-forward method.

## Problem statement

In an E-drop or effective message-drop policy, a message of the same size or nearly the same size is randomly dropped to create space for incoming messages. Small-size messages will remain for a longer time in the buffer (*Rashid et al., 2011*; *Bibi et al., 2022*). As this policy drops the messages randomly, both the larger and smaller messages have an equal probability of being dropped. There may be a chance to drop a new message that has been received recently, and this new message has less delivery probability as compared to other old messages. The main drawback of the E-drop policy is that it does not work properly for newly generated messages and has no priority for new or old messages to be dropped.

To resolve the mentioned problem of the E-drop policy, a new message drop policy is proposed which drops a message from the buffer of equal size or nearly equal size of an incoming message. The proposed policy uses the TTL metric and drops the message from the buffer with the lowest TTL. It always drops the oldest message from the buffer to create space for the incoming message. This proposed policy efficiently manages the buffer memory and provides a high chance for new messages to be delivered to their destination.

## Objectives

The main objectives are:

1. To increase the forwarding probability of new and small messages.

2. To minimize the overhead ratio of the network.

3. To decrease unnecessary messages, drop due to TTL metric.

4. To implement the proposed message drop policy using ONE simulator and evaluate its performance by comparing it with the existing policies.

### Significance of the research

The proposed message drop policy will drop messages having the lowest TTL. Different parameters have been used by different researchers, but the lowest TTL parameter has not been used in any buffer management scheme. This proposed policy drops equal size or nearly equal size messages to accommodate the incoming messages. Due to this new message drop policy, the forwarding probabilities of the new message will increase and only the messages with the lowest TTL with first in first out (FIFO) manner will be dropped to create space for incoming messages. This message drop policy will avoid buffer congestion, increase the delivery probability of small and new messages, and decrease network overhead.

The rest of the article is organized as follows: "Related Work" includes related work, the proposed scheme is discussed in "Proposed Scheme", the simulation and results are thoroughly discussed in "Simulation and Results", and in "Conclusion and Future Works" research is concluded and future work is elaborated.

## RELATED WORK

In DTNs, buffer congestion occurs as there is no end-to-end connectivity between sender and receiver all the time. There are many reasons for congestion like non-availability of end-to-end connectivity, nodes movement, limited energy, and fixed buffer memory. These are major factors that hinder communication between sender and receiver. To minimize message congestion and improve the delivery probability, different techniques have been proposed by different authors.

The researchers *Rashid et al. (2011)*, presented a message drop policy called effective drop (E-Drop). It has made the comparison with mostly forwarded (MOFO) which does not consider all the metrics of a message. Now the E-drop has been proposed by the author to minimize the network relay, overhead ratio, average latency, and hop count average and to enhance the message delivery and buffer time average. This policy selects random-size messages of equal or nearly equal size to the incoming message from the buffer and drops that message to make room for the incoming new message. As it drops the messages randomly there is a high chance of dropping a young and new message which has a low delivery probability as that of other old messages. The simulation results show that it works better in terms of delivery probability and overhead ratio than that of MOFO. It reduces the unnecessary message drop due to the selection of appropriate messages.

Researchers *Shin & Kim (2011)* proposed a buffer management policy called the Enhanced Buffer Management Policy (EBMP). This policy focuses on improving message delivery and decreasing the average message delay. To obtain these aims, it uses message properties like the number of message copies, message age, and leftover TTL to calculate the utility value of the message. With the help of these utility values, a message can easily be dropped from the buffer when the buffer overflows. The simulation results revealed that it works better in terms of delivery probability and overhead ratio as compared to other policies. The proposed policy results show that it works better while having no knowledge of network structure and size on real-world mobility trace data and synthetic data.

The researchers *Ali, Qadir & Baig (2010)* proposed a buffer management policy called size aware drop (SAD). The main goal of this message drop policy is to minimize unnecessary message drops and improve the consumption of the buffer. This policy tries to pick the appropriate size message; hence its overhead ratio is low and delivery probability is higher than that of other policies. The simulation result under random waypoint is better than MOFO, drop largest (DLA), and drop oldest (DOA). The simulation parameters of this buffer management policy are message delivery rate, overhead ratio, and Buffer Time Average (BTA). This message drop policy first selects an equal-sized message from the buffer and if not then it will select multiple small messages to accommodate the incoming new message.

The researchers *Moetesum et al. (2016)* suggested a mechanism for the avoidance of message congestion which is called a novel buffer management scheme. This policy works on two parameters *i.e.*, hop count and TTL. It uses a fixed threshold value, and the hop count value of a message is compared with that threshold value to drop a message from the buffer. The message will remain in the network if its hop count value is greater than the threshold value according to its TTL value. The transmission opportunity of low hop count messages is very high because these messages have been passed to a certain number of hops while moving to the destination. The delivery ratio will be low if low hop count messages are dropped from the buffer.

Research conducted by *Samyal & Gupta (2018)* proposed a buffer management policy to avoid buffer congestion. This policy divides the messages into three categories *i.e.*, source, relay, and destination. A separate metric has been kept dropping messages to accommodate new incoming messages. This metric automatically activates the Drop Expire Module when a certain criterion is met, and the message is dropped from the buffer. This policy will store the message in the buffer if there is enough space in the buffer and when the buffer overflows then this policy will activate another module to drop another message and create space for incoming messages.

The authors *Ayub et al. (2018)* presented a buffer management scheme called Proposed Buffer Management Policy (PBMP). This policy considers two metrics *i.e.*, TTL and number of forwards (NoF), This policy drops a message from the buffer having the lowest TTL and highest value of forwarding. The main theme of this policy is to reduce the unnecessary message drop ratio and enhance the delivery of messages. The experimental results have shown better results in terms of maximization of delivering ratio as compared to MOFO by 13.18%. The reason behind the good delivery ratio is due to these two metrics TTL and NoF.

The researchers *Kang & Chung (2020)* proposed a mechanism to evaluate the performance of different buffer management policies. This mechanism uses Opportunistic Network Environment (ONE) simulator as a simulation tool and six different simulation parameters to evaluate the performance of different message drop policies. The simulation parameters that have been used in this research work were delivery probability, latency average, hop-count average, overhead ratio, and message drop ratio. The simulation results reveal that the delivery probability of all message drop policies is 25% below by using the Spray & Wait routing protocol.

The buffer management policy proposed by *Rashid & Ayub (2023)* manages the buffer by a community metric which consists of global rank value (GRV) and local rank value (LRV). The main function of GRV is to encounter the community by any individual node that holds the message destination. Similarly, in LRV, a node connection is directly established with the destination message. A node will only drop a message from the buffer if it meets the criteria of threshold value and a message having high GRV and LRV with minimum threshold cannot be dropped from the buffer. Different locking mechanisms have been used in the message header. The experimental results show better performance than DOA, Last In First Out (LIFO), MOFO, Ndrop, Evict Shortest Life time first (SHLI), and DLA with Social Community Buffer Management (SCBM) using certain parameters by taking real-time mobility scenarios. This policy increases the delivery ratio and minimizes the overhead.

The buffer management policy by *Karami & Derakhshanfard (2023)* introduced Routing Protocol Remaining Time Destination (RPRTD) routing protocol in DTNs. This approach is based on remaining time to encounter nodes with the destination node of messages (RTD). This algorithm cleverly identifies which message should be transmitted and which message should be dropped and also identifies the sender node. This proposed method can manage buffers effectively, does not require large amounts of storage, improves the number of delivered messages, and decreases delivery delay compared to the state-of-the-art.

The approach proposed by *Sonkar, Pandey & Kumar (2023)* uses grey wolf optimization (GWO) and is utilized for bundle relaying in DTNs. This policy consists of two phases *i.e.*, deployment and relay selection. In the first phase, the GWO technique is used to find the optimal places in the network to maximize contact opportunity so that the delivery ratio can be increased with minimum delay. In the later phase, a new scheme with the name probability-based relaying scheme has been proposed which uses distance, time, and speed to calculate the probability of each encountered node. The experimental results show that it gives us better results in terms of delivery ratio if we change the number of nodes and buffer size.

The message forwarding technique introduced by *Tu et al. (2024)* investigated the social and mobility information of nodes by using historical encounter information. The Meta Meeting Mountain is used to represent the characteristics of the node, and this can be done by taking the migration degree and community relation. This scheme uses the concept of forwarding degree (FD) and with the help of this degree, a node is capable of delivering messages and improving the delivery ratio of messages. In this new forwarding scheme, different metrics *i.e.*, TTL, buffer size, and hop count are considered to increase the delivery ratio. The proposed scheme has significantly increased the delivery ratio as compared to other schemes by using different routing protocols.

The research conducted by *Ramazanzadeh & Derakhshanfard (2023)* has presented a scheme with the aim to dynamically distribute the copies of messages in the network by using an algorithm. The algorithm used in this scheme determines the number of copies in the spray phase and then forwards those messages to destinations in the forward phase using some utility value. The utility may be obtained from all those nodes which are mostly

identified. Comparison was made with some other schemes by using flooding and forwarding-based protocols and it was found that this scheme has better results as compared to those schemes.

## PROPOSED SCHEME

This research consists of a proposed message drop policy, which drops equal-size messages or almost equal sizes with the lowest TTL to create space for incoming messages. Due to this proposed message drop policy, the forwarding probability of new and small messages is increased.

### Proposed message drop policy

This proposed message drop policy works on the management of buffer memory by removing equal or almost equal-size messages from the buffer to store the new incoming message. This proposed message drop policy takes two metrics of the message *i.e.*, size and TTL to drop the appropriate message. This policy gives priority to message size and when the message size condition is fulfilled then this policy process drops the messages from the buffer having the TTL value smallest. It reduces the unnecessary message drop due to the lowest TTL metric. This proposed policy has been adopted to increase the delivery probability of new and small messages and to minimize the overhead ratio. The detailed flow sheet of the proposed message drop policy is given in Fig. 4.

In Fig. 4, first, the size of an incoming message will be calculated and then the free buffer space will be calculated using Eq. (1).

$$\text{fbs} = \text{size of buffer} - \text{size of total messages in buffer}. \tag{1}$$

If the buffer has enough space and the size of the new message is less than the free space, then that message will be stored in the buffer. If the free buffer space is less than the message size then due to buffer congestion, the message would be dropped. For the creation of space in the buffer, a threshold "Ts" value will be calculated, and this value shows the exact space required for an incoming message. The threshold value can be calculated using Eq. (2).

$$\text{Ts} = \text{IMS} - \text{fbs}. \tag{2}$$

If the buffer has some messages equal to the threshold value, then the equal size message with the lowest TTL would be dropped and the new message will be stored in the buffer. If the buffer has no equal size message as that of the threshold value, then the message in the range of threshold value and 20% of threshold value with the least large size and lowest TTL would be dropped.

## SIMULATION AND RESULTS

This section consists of the simulation parameters, performance evaluation parameters, and simulation results generated.

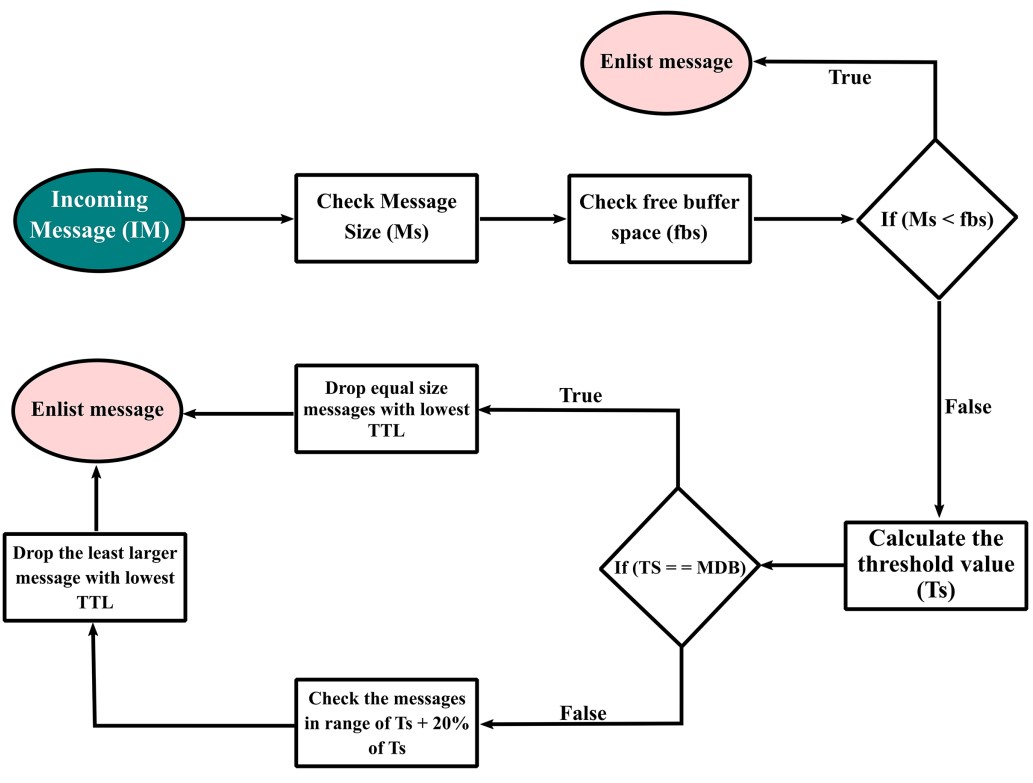

**Figure 4** **Flow chart of the proposed system.**

## Simulation parameters

The main simulation parameters of the proposed research are mentioned in Table 1. These simulation parameters are used to evaluate the performance of RAD, SAD, and DOA.

## Performance evaluation parameters

The performance of the proposed message drop policy is measured with the help of these parameters. The performance evaluation parameters which will be used in this research are as follows:

### Messages dropped

Message drop shows the number of messages dropped from the buffer during congestion. The unnecessary message drop should be minimized to increase the delivery ratio.

$$\text{M.D} = \text{Total messages sent} - \text{Total messages received.} \tag{3}$$

### Message delivery probability

The message delivery probability shows the total messages that are effectively transferred to the destination. It is the fraction of the messages sent over messages received. High delivery probability shows a maximum number of messages delivered to the destination. The message delivery probability should be maximized. The message delivery probability can be calculated using Eq. (4) (*Rashid et al., 2011*).

**Table 1 Simulation parameters.**

| Parameters | Value | | |
|---|---|---|---|
| | Set 1 | Set 2 | Set 3 |
| No of nodes | 15 | 30 | 45 |
| Movement model | Shortest path map-based movement model | | |
| Routing protocol | Epidemic | | |
| Buffer size | 3 MB | 6 MB | 9 MB |
| Group interface | Bluetooth | | |
| Transmission speed | 300 kbps | | |
| Transmission range | 15 m | | |
| Nodes speed | 1–3 m/s | | |
| TTL | 3,600 s | | |
| Message creation interval | 15–25 s | | |
| Message size | 100 KB–1 MB | | |
| Simulation area | 4,500 × 3,400 | | |
| Simulation time | 43,200 s | | |

$$\text{D.P} = \frac{\text{Messages Delivered}}{\text{Messages Created}}. \tag{4}$$

### Overhead ratio

The ratio between the messages relayed and messages delivered is called the overhead ratio. When the relayed messages are delivered in a short time and less processing is required, then the overhead ratio will be low. The main objective of buffer management is to reduce the overhead ratio. The overhead ratio can be calculated using Eq. (5) (*Ali, Qadir & Baig, 2010*; *Moetesum et al., 2016*).

$$\text{O.R} = \frac{(\text{Messages Relayed} - \text{Messages Delivered})}{\text{Messages Delivered}}. \tag{5}$$

### Buffer time average

Buffer time average is the total time used by a message in the buffer. When a message spends more time in the buffer then its forwarding chance is increased. So, the delivery probability of a message with a high buffer time average is increased. Buffer time average can be calculated using Eq. (6) (*Rashid et al., 2013b*; *Silva et al., 2017*; *Arslan et al., 2022*).

$$\text{B.T.A} = \sum_{i=1}^{n} \frac{\text{Message(i) Stay Time}}{\text{Messages in Buffer}}. \tag{6}$$

### Hop-count average

The hop-count average shows from how many nodes the message has been passed while reaching the destination. A message passing from a different number of nodes depends on the routing technique that has been adopted by the network to send the message from the

source to the destination. The hop-count average should be minimized to reach the message to the destination on time and to use the power and resources for a short period (*Sun, Liu & Wang, 2011*; *Rashid, Ayub & Abdullah, 2015*; *Abu Bakar et al., 2023*).

### Latency time average

Any type of delay in the processing of network data is called latency time average. A small delay shows a low latency time average and a large delay shows a high latency. Usually, the latency in DTN is high due to the non-availability of end-to-end connections (*Li et al., 2009*).

### Message relayed

The number of messages sent to other nodes still reaching the destination is known as messages relayed. It usually consumes resources in terms of buffer space, power, and bandwidth (*Naves, Moraes & Albuquerque, 2012*; *Harrati & Abdali, 2016*).

## Simulation results

The performance of the proposed range aware drop (RAD) is validated through a wide-range simulation experiment. The main purpose is to show that RAD enhances the delivery probability while minimizing the overhead ratio. The performance evaluation of the RAD has been done with size aware drop (SAD) and drop oldest (DOA).

The SAD policy drops equal size or least larger messages from the buffer to adjust the new incoming message. This policy selects multiple small messages to be dropped from the buffer if all the messages in the buffer have a small size as compared to incoming messages. It calculates a threshold value which tells us the exact space required for an incoming message in the buffer. By using this threshold size, a message of equal or nearly equal size to the threshold value is dropped and the new message is accommodated. It drops the messages randomly and there is no priority for the messages having equal size in the buffer. It minimizes unnecessary message drops due to the appropriate selection of messages. It maximizes the delivery probability of small and new messages and minimizes the overhead ratio.

The other message drop policy with which performance evaluation of RAD has been performed is DOA. In this message drop policy, a message of a large size in the buffer is dropped to create space for incoming messages. There is no such priority for large messages but drops when the first largest message is encountered. It does not manage the buffer memory better in case of small incoming messages.

In this section, the RAD scenario, different evaluation metrics, and experimental results are presented and analyzed.

For this research, a specific scenario has been settled. In this scenario, a set of 15, 30, and 45 nodes having buffer memory of 3, 6, and 9 MB has been taken. In these nodes, 15 nodes were sender, 30 were receiver, and 45 were relayed nodes. The routing protocol used for this proposed policy was an epidemic router which can transmit a message if the contact and buffer memory of nodes is high. For communication purposes, the SPMB model has been used. The interface among the nodes was Bluetooth and the total distance was kept at 15 m. The message creation time was 15–25 s, which is so minimum time for message

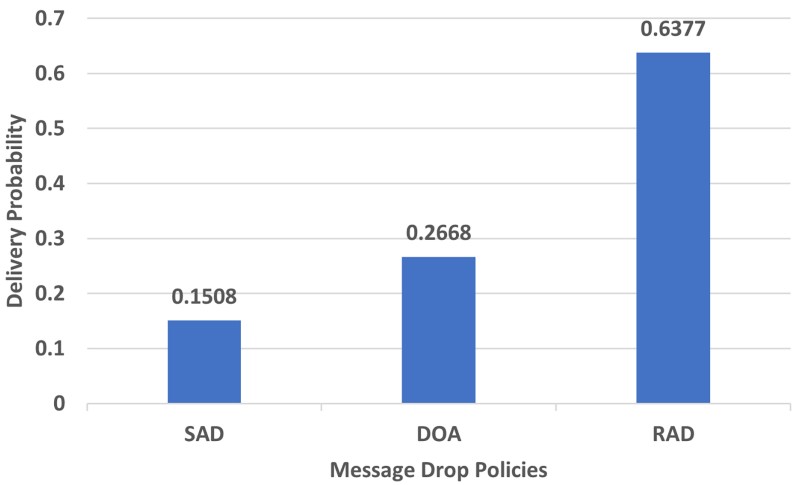

**Figure 5 Performance evaluation of SAD, DOA, and RAD in terms of delivery probability.**

creation. The TTL of the message was 60 min and the scenario run time was 43,200 s. The area selected for simulation purposes was 4,500 × 3,400.

First, the free buffer space in the buffer is checked and will store the new message if the size of the available free buffer space is greater than the incoming message. If it is not true, then appropriate space for incoming messages by assigning a threshold value will be created. This threshold value will compute the actual required free buffer space for incoming messages. Then a message of equal size or nearly equal size with the lowest TTL in FIFO order from the buffer is dropped and the new incoming message is accommodated.

### Delivery probability

Delivery probability shows how many messages have been transferred successfully to the destination. The messages should be delivered in an adequate amount of time. The performance evaluation in the case of the delivery probability of RAD, DOA, and SAD using an epidemic router is reflected in Fig. 5.

In Fig. 5, the delivery probability of SAD is 0.1508, DOA is 0.2668 and RAD is 0.6377 which shows that our proposed message drop policy significantly increases the delivery ratio of the messages. The main reason behind the increase in the delivery probability is that RAD picks the appropriate size message from the buffer with the lowest TTL to provide free space for new messages and avoid unnecessary message drops. It chooses the lowest TTL message in FIFO order from the buffer as the delivery probability of those messages is always high. Therefore, RAD performs better than SAD and DOA in terms of delivery probability due to the appropriate size and lowest TTL metric.

### Overhead ratio

The overhead ratio is the ratio between messages relayed and messages delivered. It should be minimized. The performance evaluation of RAD, DOA, and SAD in terms of overhead ratio using an epidemic router is given in Fig. 6.

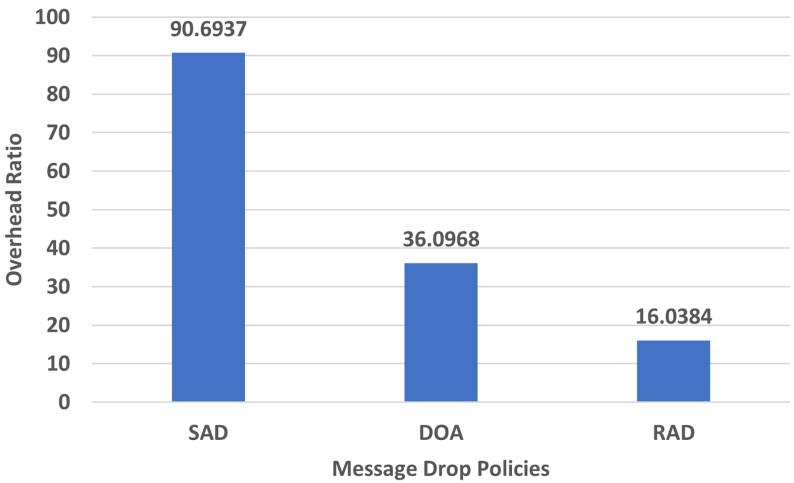

**Figure 6** **Performance evaluation of SAD, DOA, and RAD in terms of overhead ratio.**

In Fig. 6, the overhead ratio of RAD has been significantly decreased as compared to SAD and DOA. The overhead ratio increases when buffer overflows frequently occur in a congested environment. RAD minimizes the overhead ratio due to proper buffer utilization and unnecessary message drop. Due to the lowest TTL metric, it strives to drop only the required message from the buffer which can easily accommodate the new message. RAD drops old messages from the buffer as that of other policies that is why it minimizes the overhead ratio.

### Hop-count average

The hop-count average shows how many hops a message has been passed to reach the destination. An increase in hop count shows that a message has consumed more resources while reaching the destination and a minimum hop count confirms less overhead and delivery delay. The hop count average depends on the routing strategy used in particular research. The performance evaluation of the hop-count average of DOA, SAD, and RAD using an epidemic router is given in Fig. 7.

The hop count average of RAD, SAD, and DOA are shown in Fig. 7. The hop count average of RAD is less than SAD and DOA due to proper size message drop and lowest TTL. These metrics do not allow the RAD to drop unnecessary messages from the buffer when congestion frequently occurs. The decrease in the hop count average of RAD shows that it uses less amount of network resources and time to reach the destination.

### Buffer time average

The buffer time average tells us how many times a message was in the buffer. When a message spends more time in the buffer its forwarding probability increases. The performance evaluation in terms of the buffer time average of DOA, SAD, and RAD using an epidemic router is given in Fig. 8.

The buffer time average (BTA) of RAD, SAD, and DOA are shown in Fig. 8. The buffer time average of RAD is higher than SAD but a bit smaller than DOA. Due to the non-

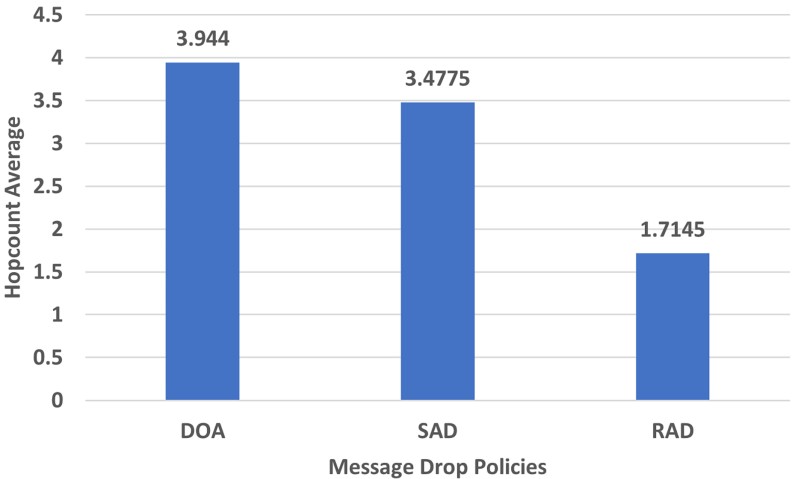

**Figure 7** **Performance evaluation of SAD, DOA, and RAD in terms of hop-count average.**

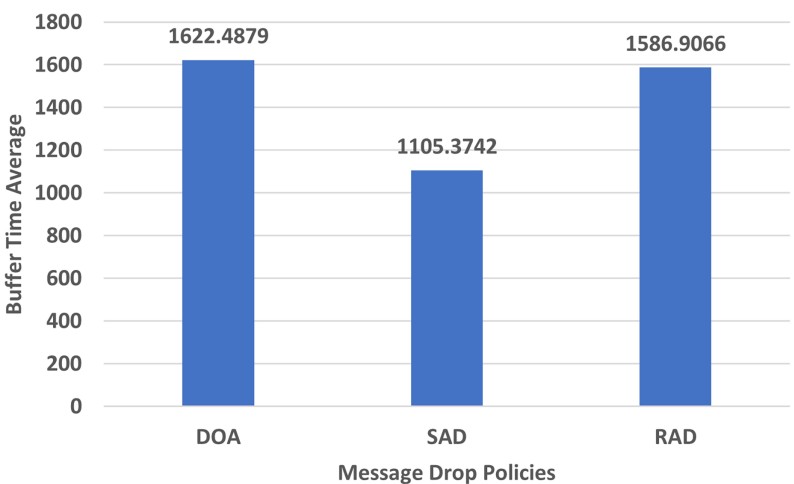

**Figure 8** **Performance evaluation of SAD, DOA, and RAD in terms of buffer time average.**

availability of connection always, the RAD has a better buffer time average than that of other policies because of the appropriate size message selection and proper utilization of buffer. The delivery probability will be high if it has a high buffer time average.

### Messages dropped

The messages dropped are the actual difference between the messages created and the messages delivered. It shows the wastage of precious resources and hence is minimized. The performance evaluation chart of messages dropped by DOA, SAD, and RAD using an epidemic router is given in Fig. 9.

The messages dropped chart in the percentage of RAD, SAD, and DOA are shown in Fig. 9. The percentage of the dropped messages of the proposed RAD policy is lower than that of SAD and DOA. The reason behind this low message drop percentage is that RAD

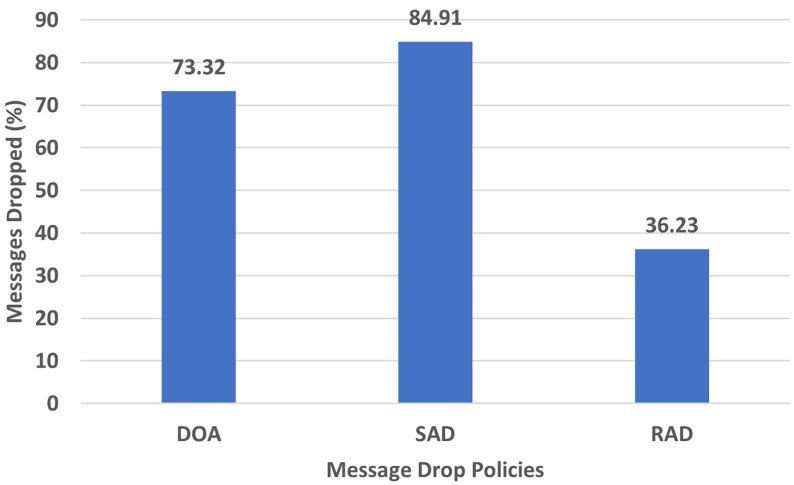

**Figure 9 Performance evaluation of SAD, DOA, and RAD in terms of messages dropped.**

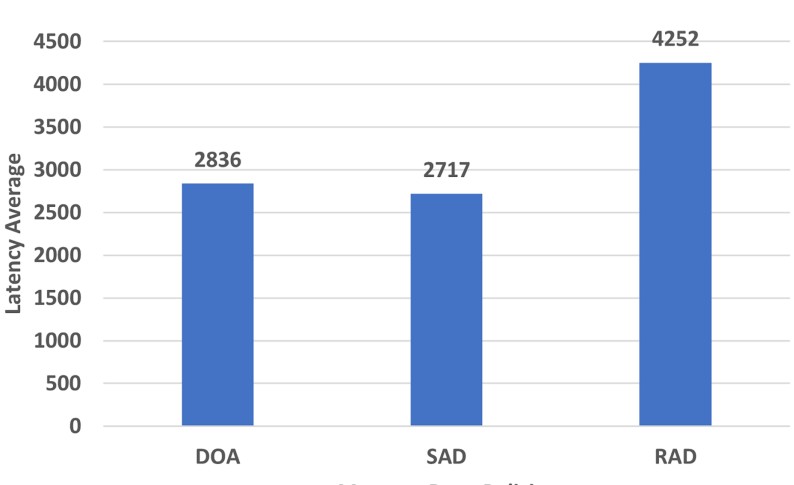

**Figure 10 Performance evaluation of SAD, DOA, and RAD in terms of latency average.**

stores messages in the buffer for a long time and drops only appropriate and lowest TTL messages to create space for incoming messages. Due to this reason, most of the messages are forwarded to the destination, and a smaller number of messages are dropped from the buffer with equal size or the least large size. So, Fig. 9 shows that RAD works better than SAD and DOA in terms of messages dropped.

### Latency average

In DTN, latency is high due to the nature of the network. Latency is typically any type of delay in the processing of network data.

Figure 10 shows the performance evaluation in terms of the latency average of RAD, SAD, and DOA. The latency average of RAD is higher than that of other policies which helps the messages to stay in the network for a long time. This long time stay of messages

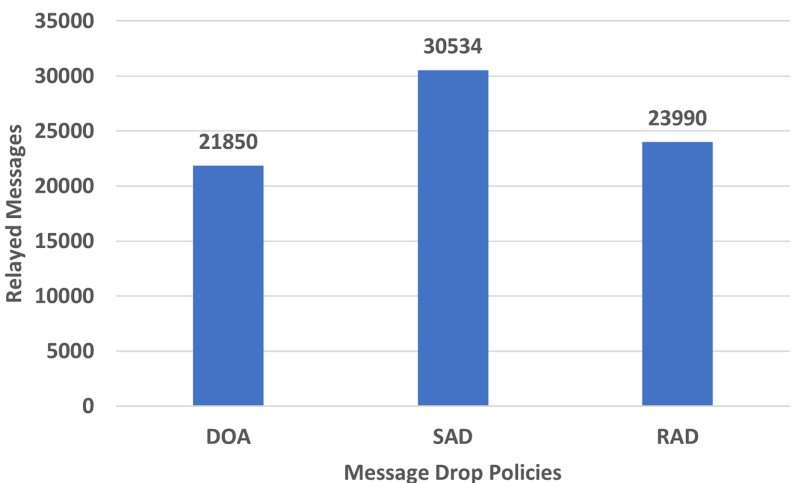

**Figure 11 Performance evaluation of SAD, DOA, and RAD in terms of relayed messages.**

helps to increase the forwarding probability of the messages and minimizes the overhead ratio. So due to no end-to-end connectivity among the nodes, the latency of all the messages in the DTNs must be high.

### Messages relayed

The relayed messages are the ratio of the messages sent to others while reaching the destination. For the storage of relayed messages, relayed nodes are used to hold the messages till the communication and contact of other nodes. The performance evaluation in terms of relayed RAD, SAD, and DOA messages using the epidemic router is given in Fig. 11.

Figure 11 shows the results of relayed messages of RAD, SAD, and DOA. The relayed helps to store a message for a long time till the establishment of a connection between sender and receiver to minimize message drop but increases the overhead. The relayed messages of RAD are less than SAD but a little bit greater than DOA as shown in Fig. 11. The increase in relayed messages in RAD as compared to DOA is due to the dropping of equal or least large messages which often takes a long time to fulfill the criteria. In DOA, the oldest message may be dropped which will always take less time to meet certain criteria.

## CONCLUSION AND FUTURE WORKS

It has been concluded in this research that appropriate message size and the lowest TTL play an important role in the efficient buffer management of DTNs. This message drop policy (RAD) uses the concept of appropriate message size and the lowest TTL for achieving the required results. RAD performs better as compared to SAD and DOA in terms of delivery probability and hence RAD significantly increases the delivery probability of messages. The RAD also minimizes the overhead ratio and hop count average significantly as compared to SAD and DOA and both should be minimized for better management of buffer as well as network. The proposed RAD policy allows the messages to stay for a huge amount of time in the buffer which enhances the delivery ratio

of the messages. Almost all the performance evaluation parameters set in the proposed policy yield better results as compared to SAD and DOA. The shortcoming of the proposed research is the scenario when the buffer has messages and all those messages have sizes smaller than the size of the incoming message which is very least likely. The proposed message drop policy such as RAD is an efficient buffer management approach. In the future, other aspects can be explored for further efficiency of buffer management in DTNs. Moreover, the interesting directions of future research in DTNs are to explore security issues, design security solutions, and use intelligent routing techniques using Artificial Intelligence (AI) and machine learning (ML) algorithms for proper message delivery and buffer management in DTNs.

### Funding

This research is funded by the Researchers Supporting Project Number (RSPD2024R947), King Saud University, Riyadh, Saudi Arabia. The funders had no role in study design, data collection and analysis, decision to publish, or preparation of the manuscript.

### Grant Disclosures

The following grant information was disclosed by the authors:
King Saud University: RSPD2024R947.

### Competing Interests

Khursheed Aurangzeb is an Academic Editor for PeerJ.

### Author Contributions

- Samiullah Khan conceived and designed the experiments, performed the experiments, performed the computation work, prepared figures and/or tables, authored or reviewed drafts of the article, and approved the final draft.
- Khalid Saeed conceived and designed the experiments, performed the experiments, analyzed the data, performed the computation work, prepared figures and/or tables, and approved the final draft.
- Muhammad Faran Majeed conceived and designed the experiments, analyzed the data, performed the computation work, prepared figures and/or tables, authored or reviewed drafts of the article, and approved the final draft.
- Khursheed Aurangzeb performed the experiments, authored or reviewed drafts of the article, and approved the final draft.
- Zahoor Ahmad conceived and designed the experiments, performed the experiments, prepared figures and/or tables, and approved the final draft.
- Muhammad Shahid Anwar analyzed the data, authored or reviewed drafts of the article, and approved the final draft.
- Piratdin Allayarov analyzed the data, authored or reviewed drafts of the article, and approved the final draft.

## Data Availability

The code is available in the Supplemental File.

## Supplemental Information

Supplemental information for this article can be found online at http://dx.doi.org/10.7717/peerj-cs.2099#supplemental-information.

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
