# Peer review of "Range aware message drop policy for delay tolerant networks"

_PeerJ Computer Science, doi:10.7717/peerj-cs.2099_

## Round 0.1 · original submission · Major Revisions

I have received two reviews, which both recommend major revisions. I would offer you the opportunity to address reviewers' comments with significant changes to your manuscript. You should pay particular attention on the novelty of your work, by utilising more up-to-date references and position your work against state-of-the-art knowledge gaps.

Reviewer 1 ·

Basic reporting

Language:

Line 31-32: The sentence structure is ambiguous, lacking clarity. Suggested revision: use more precise language, like 'continuous direct connection between sender and receiver' instead of vague terms.

Line 41-46: The sentences are short, incomplete, and fail to convey the intended message effectively.

Line 129-130: The statement about the probability of dropping new messages due to their low forwarding probability compared to older messages is incorrect. It's crucial to note that the dropping probability for new and old packets is the same, which reduces the forwarding probability of new packets.

Line 146: The term Random Way Point is more commonly written as Random Waypoint

Figures:

Figures 1, 3, and 4 are blurry and of low quality. The author should consider rendering these images at a higher resolution for clarity.

Experimental design

The DTN methodology and architecture are well explained. However, this is a well-known area, and such detailed descriptions may be redundant for an informed audience.

Validity of the findings

The paper is supported by a substantial number of references, but there's an over-reliance on certain sources, and many are outdated (more than 10 years old). This does not demonstrate a thorough, up-to-date literature review.

Merely combining two well-established techniques does not constitute significant technical advancement.

The related works section is outdated (citing works from 2011, 2010, 2016, 2018, 2020) and lacks recent literature, undermining the confidence in covering recent developments and motivations.

Additional comments

The flow chart and the algorithm (Fig 4) essentially convey the same principles. It is recommended to retain only the algorithm for brevity and clarity.

The descriptive section following the algorithm is redundant and could be omitted.

Overall, the paper's novelty is limited, combining well-known message drop techniques without providing significant insights.

The quality of writing and presentation needs significant improvement, particularly in highlighting the key contributions and innovations of the work.

Cite this review as

Reviewer 2 ·

Basic reporting

no comment

Experimental design

no comment

Validity of the findings

no comment

Additional comments

Delay Tolerant Networks is special case of intermittently Connected network which have intermittent connectivity and long delay. DTNs have scarce/limited resources such as buffer. In DTNs nodes store incoming packets/messages in persistence storage. The malicious nodes launch attacks to overflow the buffer of nodes, this increases packet loss ratios and decreases packet delivery ratios. The authors proposed a buffer management scheme to tackle this problem, which is very important and challenging research area.

The authors proposed an efficient scheme to cope this problem and thus possible improvement in this security domain (little contribution). However, the most important thing in paper is coherence and cohesion. Remember you are writing paper for other not for yourself. You must take care of presentation.
There are so many repetition and grammatical problem, i would suggest to the authors to critically review the presentation and grammar with the help of senior member/author of this paper.

The problems of this paper is too much that is why i do not want to write any more comments. I would encourage the authors to severely review this paper before submission to any other conference/journal. You must comprehensive paper for journal otherwise go for conference.

Cite this review as

---

## Round 0.2 · Minor Revisions

Please address the remaining concerns raised by the reviewers.

Reviewer 1 ·

Basic reporting

The manuscript has seen considerable improvements in readability, and I am pleased with the majority of the revisions implemented by the author. However, there remain a few areas that could benefit from further refinement to enhance the manuscript's overall quality:

1. There appears to be a typographical error in line 22 of the abstract, where the word "yellow" is used inappropriately. I suggest revising the sentence from "This leads to low delivery probability and yellow thus...." to correct this oversight.

2. While the quality of writing has notably improved, there are still some minor grammatical and typographical errors scattered throughout the manuscript. A thorough review and revision of the document are recommended to address these issues.

3. Regarding the visuals, while Figures 1 and 4 meet the expected standards, Figure 3 requires attention. The quality of this figure is compromised, particularly the legibility of smaller texts, which are difficult to decipher.

4. Using yellow for tracking changes in the document significantly hampers readability. For future submissions, I would recommend employing colors like red or blue for this purpose, as they offer better visual clarity and are easier on the eyes.

5. In Figure 4, the flow chart redundantly leads to the same decision ("enlist message") through different routes. To streamline the presentation and enhance clarity, it would be beneficial to consolidate these pathways into a single section where all routes converge on the same action. This can be accomplished with the more thoughtful organization of the flow chart to ensure clear and logical inputs and outputs.

Experimental design

The manuscript includes an algorithmic description that is not distinctly identified as an algorithm within the text. As noted in my previous review, this section is redundant, given that Figure 4 describes the same principles effectively. Consequently, I recommend that the author eliminate Section 3.3.8 or revise the pseudocode to enhance its efficiency and presentation.

Validity of the findings

I am happy with the revised results section.

Cite this review as

Reviewer 2 ·

Basic reporting

No Comment

Experimental design

No Comment

Validity of the findings

No Comment

Additional comments

This paper poses potentially novel approach for addressing Buffer Management scheme (Range Aware Drop) for Delay Tolerant Networks, thus possible improvement in the subject domain. The authors have laid out a theoretical basis for their approach and have attempted to explain in some formal way. In previous round of review the level of the text/presentation is very boring and confusing and has a lot of repetitions and grammatical mistakes. In fact, it made it impossible for me to go through the paper, as I had to make many assumptions about what they are trying to state at each point. However, on a positive note in this version the authors improve the presentation and coherence /cohesion of paper; I appreciate the efforts of authors in this version, that is why I would recommend to accept this article. I suggest to the authors write the organization of paper at the end of introduction section. Furthermore, I would suggest to the authors to organize the paper in this manner.
Section 1: Introduction, Section 2: Related Works, Section 3: Proposed Scheme, Section 4: Simulation and Results, which contain simulation parameters, evaluation parameters, simulation results and comparison with state of art works, followed by conclusion and future works.
What is the difference between E-Drop and SAD policy? I saw exactly the same definition in your paper, can you explain this.
“In the E-drop policy, a message from the buffer with an equal size or nearly equal size of incoming message would be dropped”.

“In SAD, a message of equal or nearly equal size from the buffer is dropped to make room for the new incoming message”.

The authors evaluate proposed scheme with only one routing protocol, Epidemic Router, I think this is not sufficient, you should evaluate proposed scheme with various parameters and routing protocols.

Although I saw few latest references in this version unlike a previous version, but I recommends you please add few more recent references, from 2018-2024.

Moreover, if you present the shortcoming of your own scheme in your paper, so this will be beneficial for other researchers in this domain (either in proposed scheme section and conclusion section).

Cite this review as

---

## Round 0.3 · accepted · Accept

It appears that the authors have responded to the comments and addressed the comments from the reviewers satisfactorily.

Reviewer 1 ·

Basic reporting

The manuscript is well-structured and articulates its objectives, methodology, and implications, adhering to the required standards.

Experimental design

The experimental design is robust and well-detailed, providing sufficient information to ensure the reproducibility and integrity of the methods used.

Validity of the findings

The findings are substantiated by solid data and a rigorous analytical approach, supporting the conclusions drawn and enhancing the manuscript’s validity.

Cite this review as